# Research

materials science/environmental chemistry

UVS-NZVI, adsorption, isotherm, kinetic, Pb²⁺, Cd²⁺

**Author for correspondence:**
Shengxu Luo
e-mail: shxluo@hainanu.edu.cn

†These authors are joint first authors.

This article has been edited by the Royal Society of Chemistry, including the commissioning, peer review process and editorial aspects up to the point of acceptance.

# Shell biomass material supported nano-zero valent iron to remove $Pb^{2+}$ and $Cd^{2+}$ in water

Zheng Wang[1,†], Xique Wu[2,†], Shengxu Luo[1,3], Yanshi Wang[2], Zhuang Tong[2] and Qin Deng[2]

[1]School of Science, [2]School of Chemical Engineering and Technology, and [3]Key Laboratory of Ministry of Education of Advanced Materials of Tropical Island Resources, Hainan University, Haikou 570228, People's Republic of China

SL, 0000-0002-7324-7376

Nanoscale zero-valent iron (NZVI) has a high adsorption capacity for heavy metals, but easily forms aggregates. Herein, preprocessed undulating venus shell (UVS) is used as support material to prevent NZVI from reuniting. The SEM and TEM results show that UVS had a porous layered structure and NZVI particles were evenly distributed on the UVS surface. A large number of adsorption sites on the surface of UVS-NZVI are confirmed by IR and XRD. UVS-NZVI is used for adsorption of $Pb^{2+}$ and $Cd^{2+}$ at pH = 6.00 in aqueous solution, and the experimental adsorption capacities are 29.91 and 38.99 mg g⁻¹ at optimal pH, respectively. Thermodynamic studies indicate that the adsorption of ions by UVS-NZVI is more in line with the Langmuir model when $Pb^{2+}$ or $Cd^{2+}$ existed alone. For the mixed solution of $Pb^{2+}$ and $Cd^{2+}$, only the adsorption of $Pb^{2+}$ by UVS-NZVI conforms to the Langmuir model. In addition, the maximum adsorption capacities of UVS-NZVI for $Pb^{2+}$ and $Cd^{2+}$ are 93.01 and 46.07 mg g⁻¹, respectively. Kinetic studies demonstrate that the determination coefficients ($R^2$) of the pseudo first-order kinetic model for UVS-NZVI adsorption of $Cd^{2+}$ and $Pb^{2+}$ are higher than those of the pseudo second-order kinetic model and Elovich kinetic model. Highly efficient performance for metal removal makes UVS-NZVI show potential application to heavy metal ion adsorption.

# 1. Introduction

Rapid growth of industrialization has created severe heavy metal pollution, and the release of many kinds of heavy metal ions in the environment not only affect the aquatic system but also

human health [1]. According to statistics, about 67 000 tons of Pb(II) ions and 40 000 tons of Cd(II) ions are discharged into the environment annually [2,3]. Most of these toxic and untreated ions reaching the soil, the surface and ground water, permanently exist and accumulate in the ecosystem [4]. Through the food chain, Pb(II) and Cd(II) may accumulate in internal organs of animals and humans, such as the kidneys and liver, and result in acute or chronic poisoning, causing various diseases [5]. Thus, in order to maintain ecological stability and public safety, effectively removing Pb(II) and Cd(II) ions from the environment have become an important research focus in environmental fields [6,7].

In previous studies, various conventional techniques have been developed for the treatment of heavy metal ions in wastewaters, such as chemical precipitation, membrane separation, ion exchange, adsorption and electrochemical methods [8–12]. The chemical precipitation method has been widely applied due to its simplicity and ease of operation. However, the effectiveness of treatment for wastewater with low concentration of heavy metal ions remains poor [13]. The membrane separation and ion exchange are very effective methods for removing heavy metal ions from water, but the operational costs are higher in comparison with other methods [14]. Of all the known methods, the adsorption technique has been regarded as a simple and effective tool for the removal of heavy metal ions from wastewater owing to its wide adaptability, environment-friendliness and low cost [15]. In recent years, nanoscale zero-valent iron (NZVI), an environmentally benign material, via its controllable particle size, non-toxic, high reactivity and abundant reactive surface sites, has been widely used to treat various metal ions in aqueous solutions [16,17]. However, direct use of NZVI is restricted due to its lack of stability, easy aggregation and facing difficulties in separating NZVI from treated effluents [18]. To address this issue, some researchers consider using stable materials as carriers to support NZVI, which could mitigate oxidization and aggregation of the NZVI, such as Zhang *et al*. used pillared clay as the carrier of NZVI, and Fu *et al*. used cashew nut shell to support NZVI [19–21]. Of all carriers used by previous researchers, most carriers are only considered for their stability and load capacity, but not for their surface structure and specific surface area. However, superior porous structure and high specific surface area can make the carrier material support more NZVI, hence, looking for such a stable carrier with a large specific surface area will promote the wide application of NZVI.

Recently, due to the development of fisheries, the production of shellfish has increased year by year, and the massive shells produced have been directly discarded due to its limited application, thus causing waste of shell resources and environmental pollution [1,22]. As a high yield biomass resource, if shellfish can be fully used in large quantities according to its characteristics, it can not only realize the reuse of waste, but reduce the damage to the ecological environment. Shell materials contain about 90% of inorganic substance and about 10% of organic substance, and the two components are evenly distributed [23]. If we remove the organic substance and keep the inorganic substance, it will form a porous structure and the specific surface area will increase significantly. Using the activated shell as a substrate to load nano-zero-valent iron can not only support the nano-zero-valent iron and prevent agglomeration, but also make full use of the porous structure of the shell, increase the specific surface area of the adsorbent, achieving physical adsorption and chemical adsorption to combine. Therefore, it is worth studying to load NZVI on the low-cost shells as an adsorbent and explore its adsorption mechanism for different heavy metals.

In the study, the most common undulating venus shells (UVS) in hainan were processed into porous UVS carrier by impurity removal, pulverization, sieving and activating. In the chemical activation method, although strong acid conditions can remove organic matter, it will destroy the $CaCO_3$ skeleton structure of the shell. However, the removal of organic matter under alkaline conditions is not complete. Therefore, we choose high-temperature calcination as the activation process. Taking $FeCl_2$ as the raw material, we compound UVS loaded nanoscale zero-valent iron adsorbent (UVS-NZVI) using borohydride reacting. The composition and structure of UVS-NZVI were characterized by SEM, TEM, IR, XRD and BET. Finally, the adsorption properties of this material for $Pb^{2+}$ and $Cd^{2+}$ in water were investigated and the thermodynamic and kinetic mechanism for adsorption of $Pb^{2+}$ and $Cd^{2+}$ by UVS-NZVI was elucidated. This work provides an effective application approach for shellfish biomass resources, and promotes the wider application of NZVI adsorbent.

# 2. Material and methods

## 2.1. Materials

Undulating venus shells (UVS) were obtained from Haikou Yongshun seafood processing market. Absolute ethyl alcohol, $FeCl_2 \cdot 4H_2O$ (analytical grade), $NaBH_4$ (analytical grade), NaOH (analytical

grade), $HNO_3$ (excellent grade), $Cd(NO_3)_2$ (analytical grade) and $Pb(NO_3)_2$ (analytical grade) reagents were purchased from Aladdin reagent (Shanghai) limited company. Deionized water was prepared by ET461275 ultra pure water machine (Lovibond, Germany).

## 2.2. Preparation of the undulating venus shell carrier

After washing with deionized water, the UVS were immersed in $0.01 \, mol \, l^{-1}$ $HNO_3$ solution for 1 h to remove the soluble impurities attached to the surfaces. Then, the shells were cleaned with deionized water and placed in a drying oven for 24 h. The washed and dried UVS were crushed and passed through a 140 mesh sieve, and the resulting powder was placed in a muffle furnace and calcined at 450°C for 60 min [1]. Finally, the optimized UVS carrier was obtained.

## 2.3. Preparation of the supported adsorbent (UVS-NZVI)

The supported adsorbent was prepared by the liquid-phase reduction process, and the synthesis process compared with NZVI-GAC of Li *et al*. has been simplified and improved [24]. The whole synthetic process was performed in the helium atmosphere. Firstly, the UVS (6.00 g) and $FeCl_2 \cdot 4H_2O$ (5.49 g) were placed into a three-necked open flask, adding ethanol/water solution (200 ml, 30%) and stirring for 30 min. Subsequently, the $NaBH_4$ solution (100 ml, $1.00 \, mol \, l^{-1}$) was added dropwise into the mixture, constantly stirring for 30 min. Finally, the black solid (UVS-NZVI) was isolated by suction filtration and dried. The reduction reaction is as formula (2.1) [25]

$$2Fe^{2+} + BH_4^- + 3H_2O + UVS \rightarrow 2E^0/UVS + H_2BO_3^- + 4H^+ + 2H_2. \tag{2.1}$$

## 2.4. Characterization of UVS-NZVI

The microstructures of the UVS and UVS-NZVI were imaged on S-4800 scanning electron microscopy (Hitachi, Japan) and H-9500 transmission electron microscopy (Hitachi, Japan). Fourier transform infrared (FTIR) spectra were obtained on a Tensor27 FTIR spectrometer (Bruker, Germany). X-ray diffraction (XRD) was recorded on a D/Max-2400 powder diffractometer (Rigaku, Japan), operating at 40 kV and 40 mA. The specific surface area and pore structure were measured on a specific surface area and porosity analyser (ASAP 2460 analyser, Micromeritics, USA). UVS-NZVI was degassed in vacuum and characterized by $N_2$ sorption at 77 K. The specific surface area was calculated by the Brunauer–Emmett–Teller (BET) method. Pore diameter were calculated by Barrett–Joyner–Halenda (BJH) method [26,27].

## 2.5. Calculation method for adsorption amount

The total concentrations of $Cd^{2+}$ and $Pb^{2+}$ were determined by a flame atomic absorption spectrophotometer (TAS-990, Purkinje General, Beijing). The adsorption amount of $Cd^{2+}$ and $Pb^{2+}$ in aqueous solution by the UVS-NZVI is calculated using the formula (2.2)

$$Q = \frac{(C_0 - C) * V}{m}, \tag{2.2}$$

where $Q$ ($mg \, g^{-1}$) was the amount of adsorption, $C_0$ ($mg \, l^{-1}$) was the initial concentration of $Cd^{2+}$ and $Pb^{2+}$ before adsorption and $C$ ($mg \, l^{-1}$) was the concentration of $Cd^{2+}$ and $Pb^{2+}$ after adsorption. $V$ ($l$) was the volume of the solution, while $m$ ($g$) was the mass of the UVS-NZVI.

Prepare lead standard series solutions with mass concentrations of 5, 10, 20, 30 and $40 \, mg \, l^{-1}$ with $0.15 \, mol \, l^{-1}$ dilute nitric acid solution. Measure the absorbance of the standard series of solutions by flame atomic absorption method; draw the standard curve of $A_{Pb}$ (absorbance)–$C_{Pb}$ (mass concentration of $Pb^{2+}$). Measurement conditions: gas flow rate $1500 \, ml \, min^{-1}$, height 5.00 mm, position 0.50 mm, working lamp current 2.00 mA, spectral bandwidth 0.40 nm and negative high voltage 300 V.

The standard curve of $Pb^{2+}$ is shown in figure 1, the standard curve equation: $y = 0.00906 + 0.02038x$, $R^2 = 0.99983$. Within the range of $Pb^{2+}$ mass concentration of $5–40 \, mg \, l^{-1}$, the linear relationship is good.

Prepare cadmium standard series solutions with mass concentrations of 0, 1, 2, 3, 4 and $5 \, mg \, l^{-1}$ with $0.15 \, mol \, l^{-1}$ dilute nitric acid solution. Measure the absorbance of the standard series of solutions by flame atomic absorption method; draw the standard curve of $A_{Cd}$ (absorbance)–$C_{Cd}$ (mass concentration

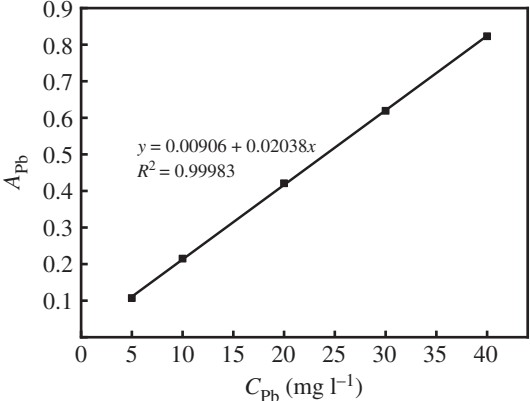

**Figure 1.** Standard curve for determining $Pb^{2+}$ content.

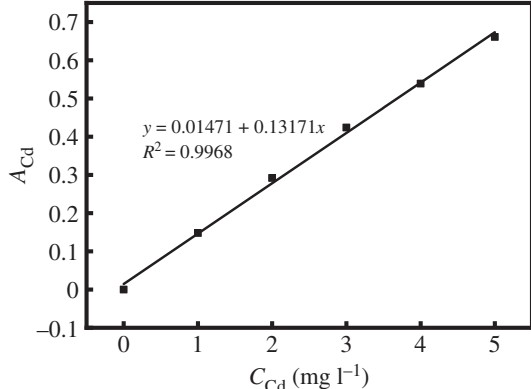

**Figure 2.** Standard curve for determining $Cd^{2+}$ content.

of $Cd^{2+}$). Measurement conditions: gas flow rate 1300 ml min$^{-1}$, height 5.00 mm, position 0.50 mm, working lamp current 2.00 mA, spectral bandwidth 0.40 nm and negative high voltage 300 V.

The standard curve of $Cd^{2+}$ is shown in figure 2, the standard curve equation: $y = 0.01471 + 0.13171x$, $R^2 = 0.9968$. Within the range of $Cd^{2+}$ mass concentration of 0–5 mg l$^{-1}$, the linear relationship is good.

## 2.6. Adsorbent dosage experiment

Six samples of 100 ml 20 mg l$^{-1}$ $Pb^{2+}$ solutions were prepared with the pH 6.00, and the adsorbent dosage set to 0.01 g, 0.02 g, 0.03 g, 0.04 g, 0.05 g and 0.06 g, respectively. The mixture was shaken at 30°C for 2 h. After filtration, the filtrate was measured for absorbance, the average value was determined with three parallel measurements, and the adsorption amount was calculated using formula (2.2).

$Cd^{2+}$ solution was used for the above experiment, and other experimental conditions were the same.

## 2.7. PH gradient setting experiment

Six samples of 100 ml $Pb^{2+}$ solutions were prepared with a concentration of 20 mg l$^{-1}$. After adjusting the value of pH from 4.00 to 6.50 (at 30°C, the solubility product of $Pb(OH)_2$ is about $1 \times 10^{-17}$; according to the concentration of $Pb^{2+}$ in the solution, calculations show that when the pH of the solution is slightly greater than 7.00, $Pb^{2+}$ begins to precipitate), 0.04 g of UVS-NZVI was added into each solution. Then, the mixture was shaken at 30°C for 2 h. After filtration, the filtrate was measured for absorbance, the average value was determined with three parallel measurements, and the adsorption amount was calculated using formula (2.2).

Nine samples of 100 ml $Cd^{2+}$ solutions were prepared with a concentration of 20 mg l$^{-1}$. The value of pH was adjusted from 4.00 to 8.00 and 0.04 g of UVS-NZVI was added into each solution. The subsequent processes were the same as $Pb^{2+}$ solutions.

The pH of each group of solutions was adjusted with dilute solutions of NaOH and $HNO_3$ by pH meter (FE28-Standard, Mettler Toledo, Switzerland).

## 2.8. Coexistent ion experiment

Seven samples of 100 ml 20 mg $l^{-1}$ $Pb^{2+}$ solutions and seven samples of 100 ml 20 mg $l^{-1}$ $Cd^{2+}$ solutions were prepared with the pH 6.00, and the $Cu^{2+}$ concentration set to 0, 2, 4, 6, 8, 10 and 12 mg $l^{-1}$, respectively. Of UVS-NZVI, 0.04 g was added into each solution. Then, the mixture was shaken at 30°C for 2 h. After filtration, the filtrate was measured for absorbance, the average value was determined with three parallel measurements, and the adsorption amount was calculated using formula (2.2).

Change $Cu^{2+}$ to $Ni^{2+}$ in the above process, and other experimental conditions are the same.

## 2.9. Adsorption thermodynamic experiment

Seven samples of 100 ml $Pb^{2+}$ solutions and seven samples of 100 ml $Cd^{2+}$ solutions were prepared, and the concentrations were 5, 10, 20, 30, 50, 75 and 100 mg $l^{-1}$, respectively. Then 0.04 g of UVS-NZVI was added to each solution and the mixture was shaken at 30°C for 2 h. After filtration, the filtrate was measured for absorbance, and the average value was determined with three parallel measurements. The adsorption amount was calculated using formula (2.2).

Also, seven kinds of different concentrations of mixed solutions of $Pb^{2+}$ and $Cd^{2+}$ were prepared and adjusted to pH of 6.00. The following steps are consistent with the above experiment. The Langmuir and Freundlich adsorption isotherm models were adopted to evaluate the isotherm experiment data.

Langmuir and Freundlich adsorption isotherm models were given as formulae (2.1) and (2.2) [28,29]

$$q_e = \frac{q_{max} b C_e}{1 + b C_e} \tag{2.3}$$

and

$$q_e = K_f C_e^{n_f}, \tag{2.4}$$

where $q_e$ (mg $g^{-1}$) is the equilibrium adsorption capacity, $q_{max}$ (mg $g^{-1}$) is the maximum adsorption capacity, $C_e$ (mg $g^{-1}$) is the equilibrium concentration, $b$ is the constant, $K_f$ is the Freundlich constant, and $n_f$ is the concentration index.

Five samples of 100 ml 20 mg $l^{-1}$ $Pb^{2+}$ solutions and five samples of 100 ml 20 mg $l^{-1}$ $Cd^{2+}$ solutions were prepared. Then 0.04 g of UVS-NZVI was added to each solution and the mixture was shaken at 20, 30, 40, 50 and 60°C for 2 h, respectively. After filtration, the filtrate was measured for absorbance, and the average value was determined with three parallel measurements. The adsorption amount was calculated using formula (2.2).

The Van't Hoff equation was used to fit the adsorption capacity at different temperatures to obtain the thermodynamic parameters $\Delta H$ and $\Delta S$ of $Pb^{2+}$ and $Cd^{2+}$. Calculate $\Delta G$ at different temperatures by Gibbs equation. Van't Hoff and Gibbs equations were given as formula (2.5) and (2.6) [30,31]

$$\ln K_C = -\left(\frac{\Delta H}{R}\right)\frac{1}{T} + \frac{\Delta S}{R}, \tag{2.5}$$

$$\Delta G = \Delta H - T\Delta S \tag{2.6}$$

and

$$K_C = \frac{C_s}{C_e}, \tag{2.7}$$

where $C_s$ is the concentration of the solid surface at the adsorption equilibrium and $C_e$ is the concentration in the solution at the adsorption equilibrium.

## 2.10. Kinetic adsorption experiment

A 50 mg $l^{-1}$ $Pb^{2+}$ solution and 50 mg $l^{-1}$ $Cd^{2+}$ solution were prepared. Then 0.02 g of UVS-NZVI was added to 100 ml of $Pb^{2+}$ or $Cd^{2+}$ solution and the mixture was shaken at 30°C. Determining the concentration of $Pb^{2+}$ and $Cd^{2+}$ every 5 min within the adsorption time of 5–60 min, the adsorption amount was calculated using formula (2.2). The pseudo first-order, pseudo second-order and Elovich kinetic models were used to determine the rate of the adsorption process.

Pseudo first-order and pseudo second-order kinetics models were given as formula (2.8) and (2.9) [32,33]

$$q_t = q_e(1 - \exp(-k_1 t)) \tag{2.8}$$

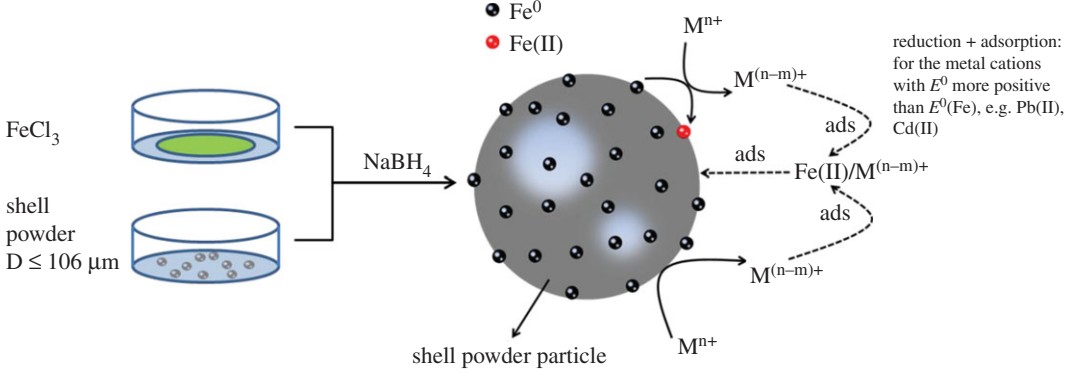

**Figure 3.** Preparation and adsorption processes of UVS-NZVI.

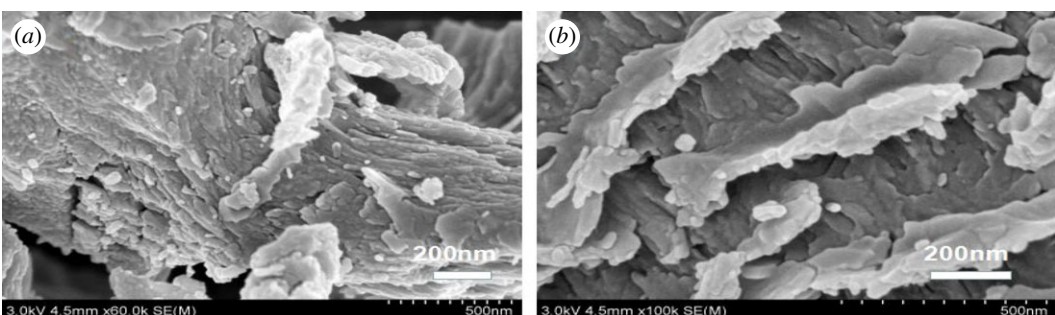

**Figure 4.** SEM images of UVS.

and

$$q_t = \frac{q_e^2 k_2 t}{1 + q_e k_2 t},\tag{2.9}$$

where $q_t$ (mg g$^{-1}$) is the adsorption capacity at time $t$ (min), and $q_e$ (mg g$^{-1}$) is the equilibrium adsorption capacity. And $k_1$ (min$^{-1}$) was the pseudo first-order kinetic rate constant, and $k_2$ (g min mg$^{-1}$) was the pseudo second-order kinetic rate constant.

Elovich kinetic model was given as formula (2.10) [34]

$$q_t = (1 + \beta_E)\ln(1 + \alpha_E \beta_E t),\tag{2.10}$$

where $\beta_E$ (g mg$^{-1}$) is the desorption constant related to the activation energy of chemisorption and $\alpha_E$ (mg (g min)$^{-1}$) is the initial adsorption rate.

## 2.11. Recyclability of UVS-NZVI

The reusability of UVS-NZVI was tested by repeated Pb$^{2+}$ and Cd$^{2+}$ adsorption and desorption cycles for five consecutive cycles. First, 100 ml of 50 mg l$^{-1}$ Pb$^{2+}$ or Cd$^{2+}$ solution was prepared. Then 0.02 g of adsorbent was added. The mixture was shaken at 30°C for 2 h. The desorption process was in 30 ml ultrapure water for 10 h.

The filtrate during each cycle was measured for absorbance. The average value was determined with three parallel measurements, and the adsorption amount was calculated using formula (2.2).

The preparation and adsorption processes of UVS-NZVI are shown in figure 3.

# 3. Results and discussion

## 3.1. Characterization of adsorbents

The porous lamellar structure of UVS is shown in figure 4a,b. More channels inside the UVS might be opened due to the acidic conditions in the preparation process, which increased the pore volume and

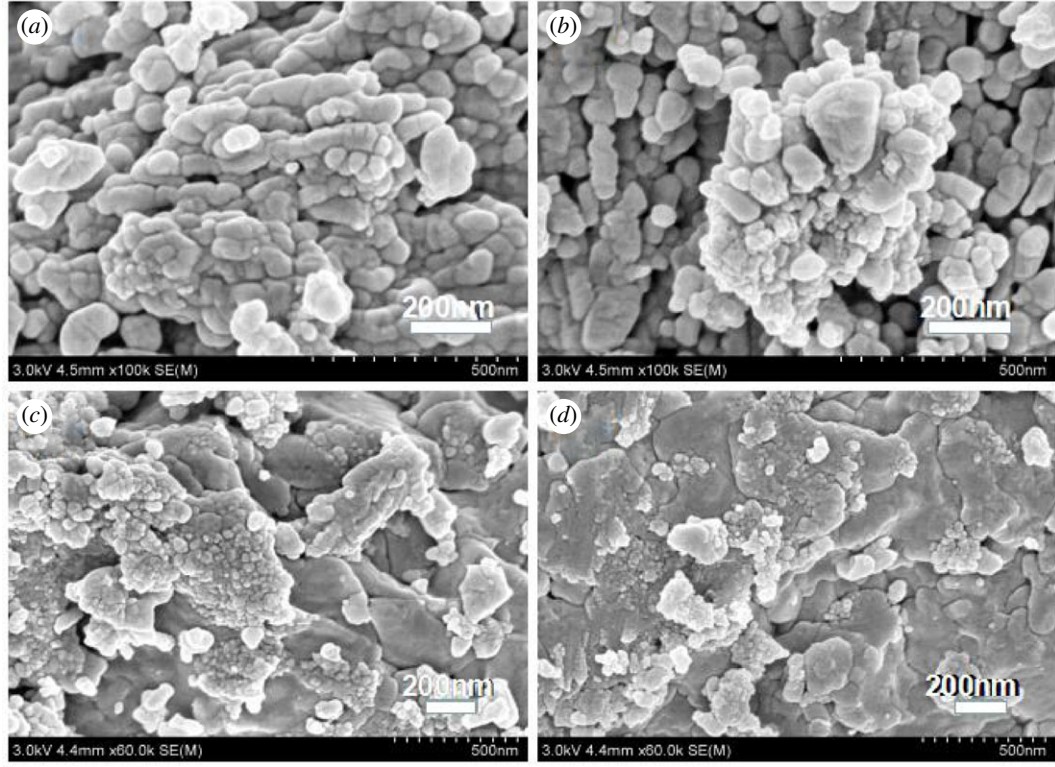

**Figure 5.** SEM images of NZVI (*a,b*) and UVS-NZVI (*c,d*).

internal specific surface area of UVS [35]. The SEM images of NZVI and UVS-NZVI are presented in figure 5. From figure 5*a,b* the NZVI particles can be seen clearly and had an average diameter of 60 nm. However, these particles are aggregated due to lack of support material [36]. According to figure 5*c,d*, NZVI particles are uniformly dispersed on the UVS surface without obvious accumulation, and the size of particles is relatively small, indicating that NZVI loaded onto the UVS successfully [37].

The structural information provided by TEM is in agreement with SEM analysis. From figure 6*a,b*, massive NVZI particles are aggregated, while in figure 6*c,d*, flocculation (NZVI or derivatives of Fe$^0$) is distributed around the UVS structure evenly [38].

The XRD patterns of UVS and UVS-NZVI are shown in figure 7. For UVS (*a*), the characteristic peaks are attributed to aragonite CaCO$_3$ and calcite CaCO$_3$, and most of the characteristic peaks are calcite CaCO$_3$ [39]. For UVS-NZVI (*b*), except for the $2\theta = 57.48°$ attributed to calcite CaCO$_3$ characteristic peaks, the rest are characteristic peaks of Fe and its compounds. As shown, the strong peaks at 44.81° and 50.93° are present on UVS-NZVI composite, corresponding to Fe$^0$ [40]. In addition, some minor peaks of Fe compounds, for example, Fe$_2$O$_3$ (55.23°) and Fe(OH)$_3$ (64.01°), are also observed, which might result from oxidation and corrosion during the synthesis and drying process [41,42].

Figure 8 exhibits FTIR of the UVS and UVS-NZVI. For UVS (*a*), the 710.58, 860.18 cm$^{-1}$, etc. are the characteristic peaks of CO$_3^{2-}$, indicating that the main component of UVS is CaCO$_3$ [43]. The peak at 860.18 cm$^{-1}$ is blue shifted by 5.68 cm$^{-1}$ compared with the peak at 854.50 cm$^{-1}$ of aragonite CaCO$_3$, which shows that UVS also contains a small amount of calcite CaCO$_3$, and the remaining peaks are essentially identical with those of aragonite CaCO$_3$. Based on the FTIR spectrum of UVS, there are no other functional groups in addition to CO$_3^{2-}$ [39,44]. Therefore, calcium carbonate did not decompose at 450°C in this experiment.

After UVS was loaded with NZVI (*b*), various peaks of iron compounds are all obvious. The peak at 3353.91 cm$^{-1}$ can be attributed to O–H association with Fe or its oxyhydroxide [45]. The peaks at 1276.71 and 1344.56 cm$^{-1}$ can be attributed to bands associated with Fe$_3$O$_4$, Fe$_2$O$_3$ and FeOOH formation surrounding Fe$^0$, as well as ethanol added in synthesizing the adsorbents [46]. In addition, 1130.26, 946.31 and 823.57 cm$^{-1}$ are the characteristic peaks of –OH in Fe(OH)$_3$, and 618.56 cm$^{-1}$ is the characteristic peak of Fe$_2$O$_3$ [47,48]. The FTIR spectrum indicates the slight oxidation of Fe$^0$, and the abundant oxygen functional groups could act as available adsorption sites, which could provide free pairs of electrons to interact with the empty orbital of metal ions (Pb$^{2+}$ and Cd$^{2+}$). This characterization result is basically consistent with the XRD, indicating the existence of zero-valent iron on the surface of UVS.

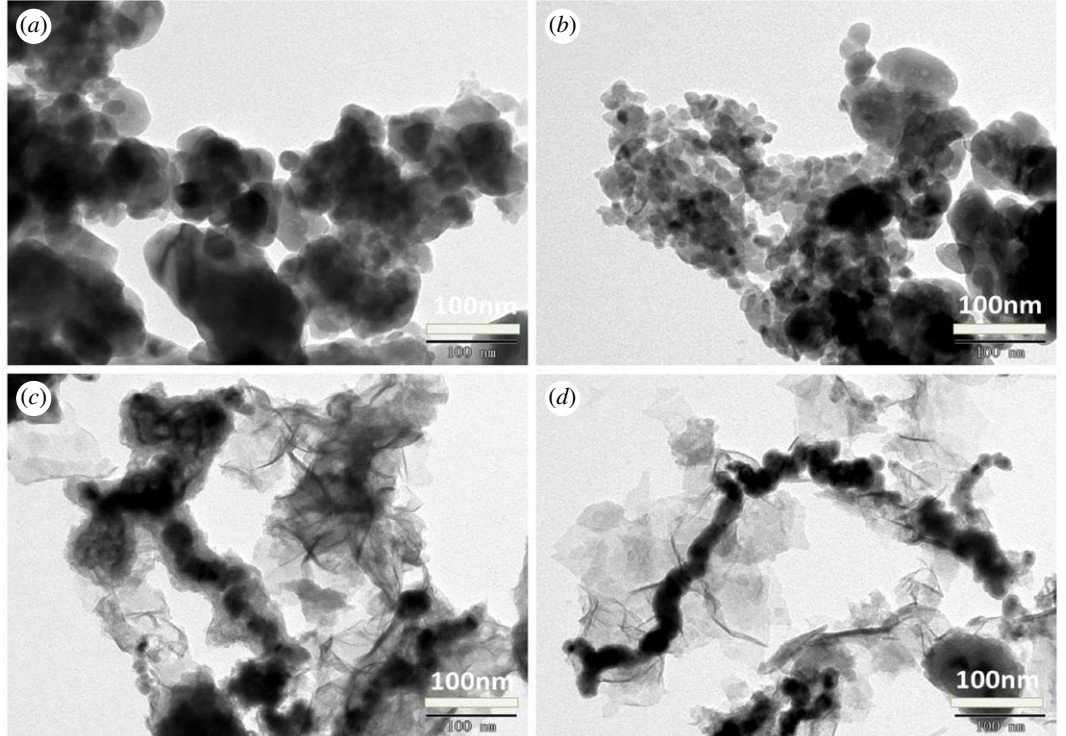

**Figure 6.** TEM images of NZVI (*a,b*) and UVS-NZVI (*c,d*).

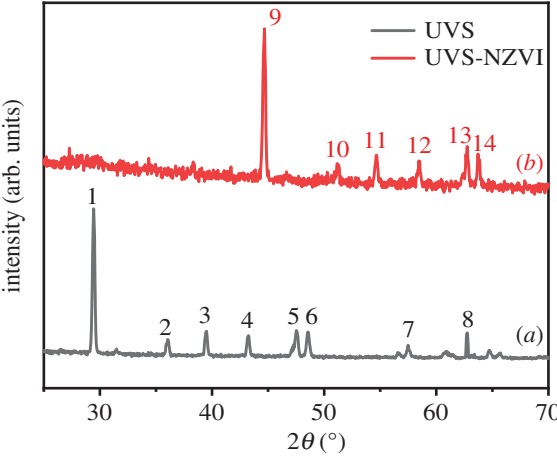

**Figure 7.** XRD pattern of UVS (*a*) and UVS-NZVI (*b*). 1–3, 5, 7, 8, 12, 13 = calcite-type $CaCO_3$; 4, 6 = aragonite-type $CaCO_3$; 9, 10 = $Fe^0$; 11 = $Fe_2O_3$; 14 = $Fe(OH)_3$.

Nitrogen sorption isotherm and pore size distribution are shown in figures 9 and 10. The specific surface area and pore diameter of UVS-NZVI calculated by BET method and BJH method are $83.77\,m^2\,g^{-1}$ and $3.54\,nm$, respectively, which indicates that the porous structure of shell material makes nano-zero-valent iron have a larger contact area with the solution, and the material may have higher adsorption performance.

## 3.2. Adsorption studies

### 3.2.1. Influence of adsorbent dosage on the adsorption of $Pb^{2+}$ and $Cd^{2+}$ ions

The effects of different adsorbent dosage on the adsorption of $Pb^{2+}$ and $Cd^{2+}$ by UVS-NZVI are shown in figure 11. For $Pb^{2+}$ and $Cd^{2+}$, when the amount of adsorbent is less than 0.04 g, the adsorption capacity

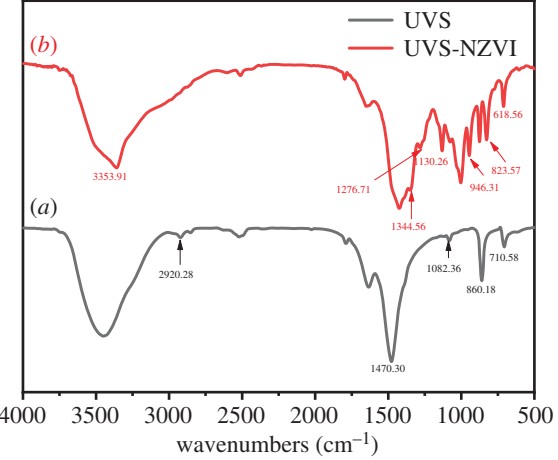

**Figure 8.** FTIR spectra of UVS (*a*) and UVS-NZVI (*b*).

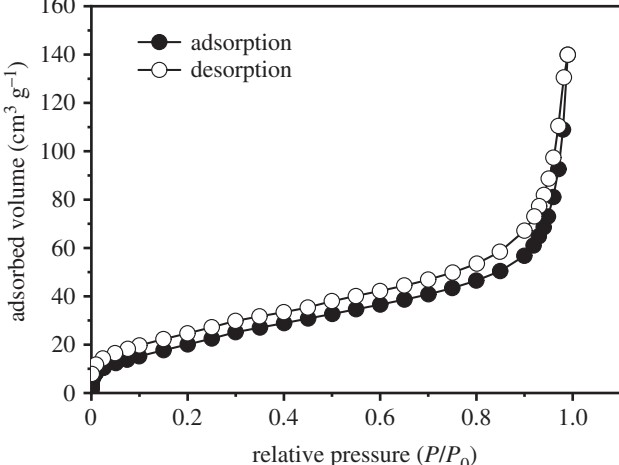

**Figure 9.** Nitrogen sorption isotherm of the sample recorded at 77 K.

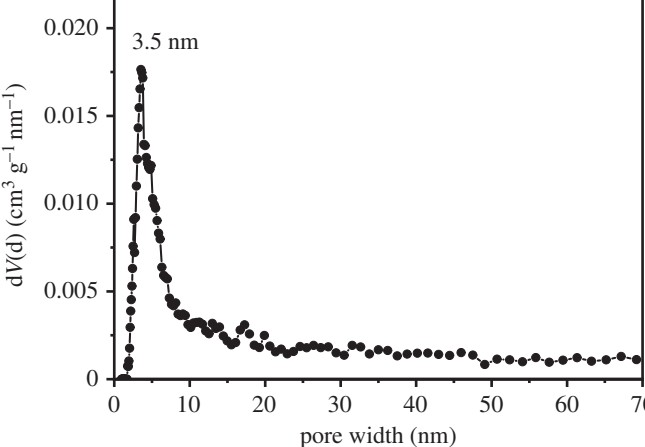

**Figure 10.** Pore size distribution.

remains almost unchanged. When the amount of adsorbent is greater than 0.04 g, the adsorption capacity drops significantly. This may be due to the ion concentration being too small, the added adsorbent has not reached its optimal adsorption effect. Therefore, in order to achieve the best removal effect and the best utilization of the adsorbent, the amount of adsorbent in subsequent experiments will be 0.04 g.

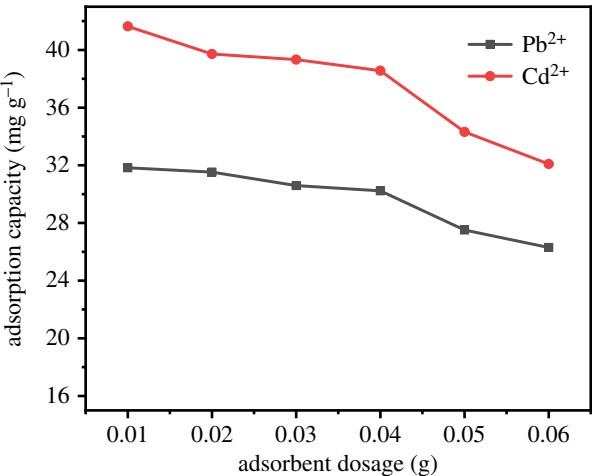

**Figure 11.** The effect of adsorbent dosage on adsorption of $Pb^{2+}$ and $Cd^{2+}$.

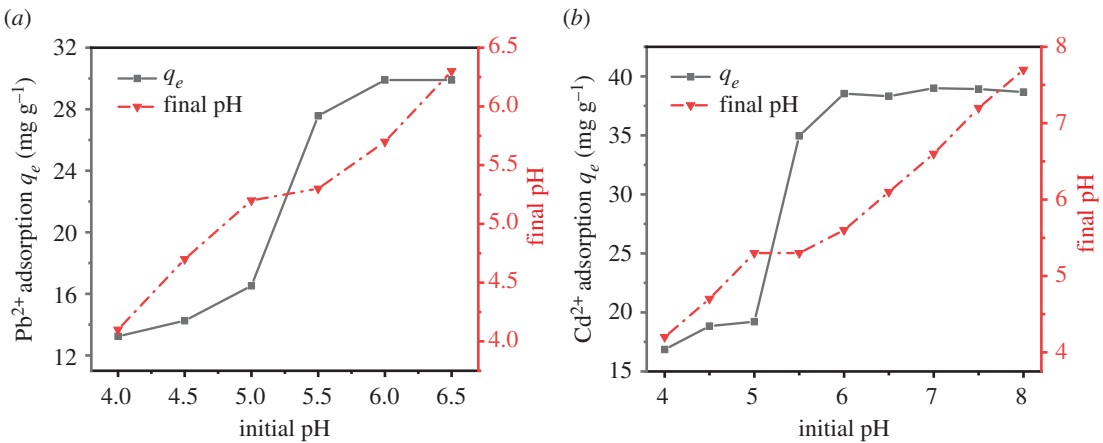

**Figure 12.** The effect of pH on adsorption of $Pb^{2+}$ (a) and $Cd^{2+}$ (b).

### 3.2.2. Influence of solution pH on the adsorption of $Pb^{2+}$ and $Cd^{2+}$ ions

According to figure 12, the pH of the solution significantly affected the adsorption of UVS-NZVI to heavy metal ions.

In strong acidic conditions, $Fe^0$ was corroded to some extent, which made UVS-NZVI poor adsorption effect on metal ions. As a result, within $4.00 < pH < 5.50$, the equilibrium adsorption capacity was less than 20 mg g$^{-1}$, whether it is $Pb^{2+}$ or $Cd^{2+}$, and the pH increased significantly after adsorption. Within $pH \geq 5.50$, adsorption performance improved obviously.

In the adsorption of $Pb^{2+}$ by UVS-NZVI, compared with $Fe^{2+}/Fe^0$, $Pb^{2+}/Pb^0$ has the higher electrode potential. Therefore, $Pb^{2+}$ can be reduced to $Pb^0$ by $Fe^0$ and the $Pb^0$ was continually adsorbed onto the iron oxide layer. Within $pH \geq 5.50$, the chemical equation of the reaction is as formula (3.1) [6,49]

$$Pb^{2+} + Fe^0 + 2OH^- \rightarrow Pb^0 + Fe(OH)_2. \tag{3.1}$$

According to the chemical equation, increasing $OH^-$ concentration is conducive to the reaction, and the adsorption process reduces the pH of the solution. As a result, for $Pb^{2+}$ (figure 12a), the equilibrium adsorption amount increased with increasing pH in the pH range of 5.5–6.5. The adsorption effect of UVS-NZVI for $Pb^{2+}$ was optimal at $pH = 6.00$, and the pH of each solution decreased after adsorption.

For $Cd^{2+}$, due to the ion exchange between $Cd^{2+}$ and $H^+$ on the surface of UVS-NZVI, pH of the solution decreased [50]. In addition, $Cd^{2+}$ could coprecipitate with $Fe^{2+}$ to form metal hydroxides over the external surfaces of adsorbents. Within $pH \geq 5.50$, the chemical equation of the reaction is as formula (3.2) [51,52]

$$nCd^{2+} + (1-n)Fe^{2+} + 2OH^- \rightarrow Cd_nFe_{(1-n)}(OH)_2. \tag{3.2}$$

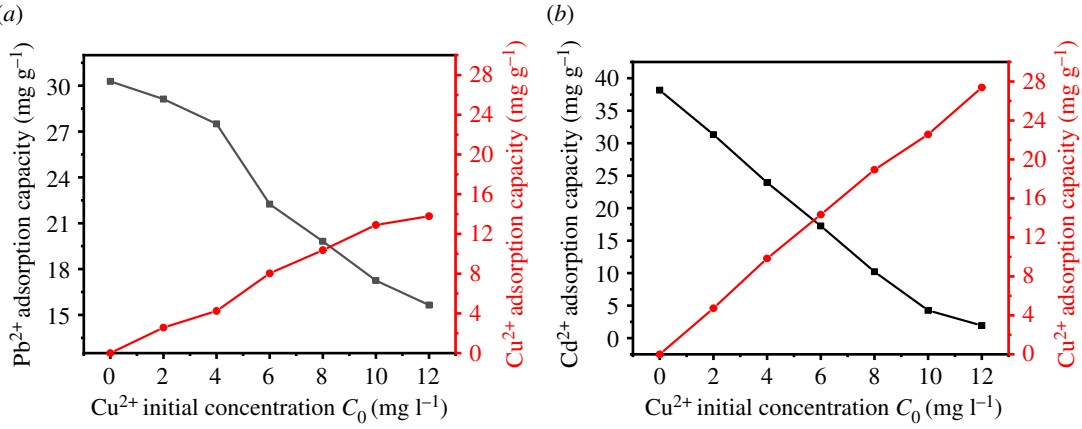

**Figure 13.** The influence of $Cu^{2+}$ concentration on the adsorption of $Pb^{2+}$ and $Cd^{2+}$.

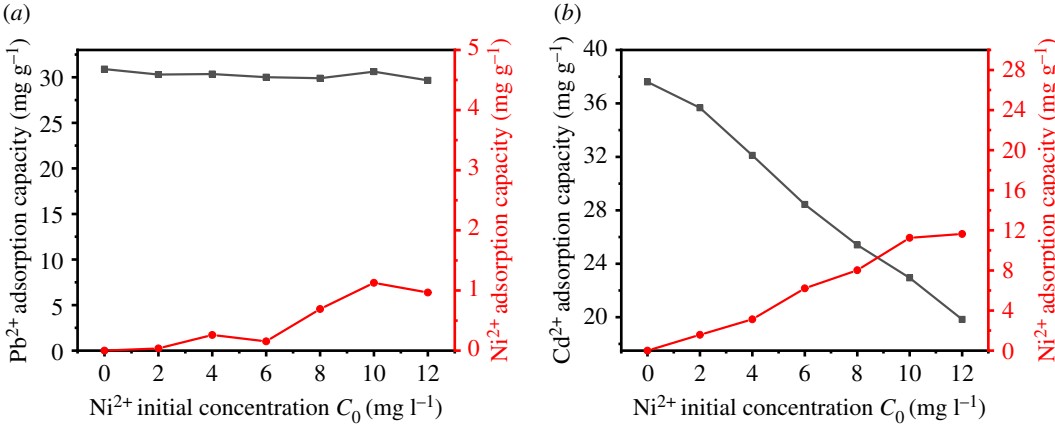

**Figure 14.** The influence of $Ni^{2+}$ concentration on the adsorption of $Pb^{2+}$ and $Cd^{2+}$.

Therefore, increasing pH facilitated the adsorption of $Cd^{2+}$ by UVS-NZVI (figure 12b). The equilibrium adsorption amount first raised and then stayed the same approximately in the pH range of 5.50–8.00. The adsorption effect of UVS-NZVI for $Cd^{2+}$ was optimal in pH ≥ 6.00.

Due to the acidic environment, the $Fe^0$ reacted with $H^+$, which limited its adsorption effectiveness. As a result, the optimal pH for the adsorption by UVS-NZVI is selected as pH = 6.00 in subsequent experiments.

### 3.2.3. Influence of coexisting ion on the adsorption of $Pb^{2+}$ and $Cd^{2+}$ ions

The effects of different concentrations of $Cu^{2+}$ and $Ni^{2+}$ on the adsorption of $Pb^{2+}$ and $Cd^{2+}$ by UVS-NZVI are shown in figures 13 and 14.

According to figure 13, the presence of $Cu^{2+}$ has a great influence on the adsorption of $Pb^{2+}$ and $Cd^{2+}$. Since the oxidability of $Cu^{2+}$ is similar to that of $Pb^{2+}$ and slightly greater than that, as the concentration of $Cu^{2+}$ increases, the adsorption performance of UVS-NZVI for $Pb^{2+}$ decreases to a certain extent. The oxidability of $Cu^{2+}$ is much greater than that of $Cd^{2+}$, so as the concentration of $Cu^{2+}$ increases, the adsorption performance of UVS-NZVI on $Pb^{2+}$ decreases significantly, and when the concentration of $Cu^{2+}$ reaches 12 mg $l^{-1}$, the adsorption capacity of $Cd^{2+}$ is only 1.93 mg $g^{-1}$, which shows that $Cd^{2+}$ is hardly adsorbed.

The effects of $Ni^{2+}$ on the adsorption of $Pb^{2+}$ and $Cd^{2+}$ by UVS-NZVI are shown in figure 14. Since the oxidability of $Pb^{2+}$ is much greater than that of $Ni^{2+}$, with the increase of $Ni^{2+}$ concentration, the adsorption capacity of $Cd^{2+}$ hardly changes, and there is almost no adsorption competition between the two ions. The oxidability of $Ni^{2+}$ is close to that of $Cd^{2+}$. As the concentration of $Ni^{2+}$ increases, the adsorption capacity of $Cd^{2+}$ decreases to a certain extent, but UVS-NZVI has still a significant adsorption capacity for $Cd^{2+}$.

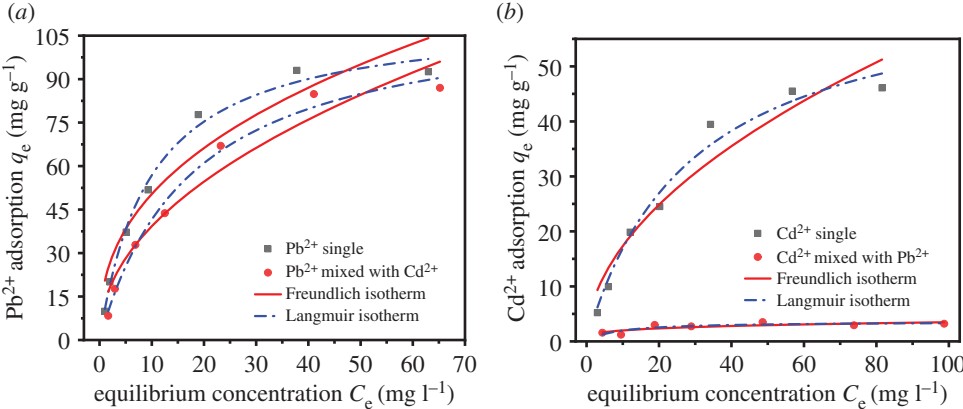

**Figure 15.** Adsorption isotherm model for $Pb^{2+}$ (*a*) and $Cd^{2+}$ (*b*) on UVS-NZVI.

**Table 1.** Langmuir and Freundlich adsorption isotherm parameters for $Cd^{2+}$ and $Pb^{2+}$ on UVS-NZVI.

| metal ions | adsorption types | Langmuir | | | Freundlich | | |
|---|---|---|---|---|---|---|---|
| | | $q_{max}$ (mg g$^{-1}$) | $b$ (l mg$^{-1}$) | $R^2$ | $K_f$ | $n_f$ | $R^2$ |
| $Pb^{2+}$ | single | 111.88 | 0.103 | 0.9911 | 20.273 | 0.395 | 0.9234 |
| | mixed with $Cd^{2+}$ | 114.928 | 0.057 | 0.9909 | 13.08 | 0.477 | 0.9499 |
| $Cd^{2+}$ | single | 66.016 | 0.035 | 0.9797 | 5.307 | 0.515 | 0.9344 |
| | mixed with $Pb^{2+}$ | 3.501 | — | — | — | — | — |

### 3.2.4. Adsorption thermodynamic analysis

The parameters and fitted plots of Langmuir and Freundlich adsorption isotherm models are listed in table 1 and figure 15*a,b*. The maximum adsorption capacity of UVS-NZVI for $Pb^{2+}$ in two different situations was 93.01 and 86.99 mg g$^{-1}$ (figure 15*a*), respectively. In the solution containing only $Pb^{2+}$, based on the higher determination coefficient ($R^2$), the $Pb^{2+}$ adsorption by UVS-NZVI was more in line with the Langmuir model, which indicates that the adsorption sites on the adsorbent are homogeneously distributed with a monolayer coverage of adsorption products [45,53,54]. The adsorption capacity of UVS-NZVI to $Pb^{2+}$ was hardly affected by the coexisting $Cd^{2+}$ in mixed solution and the adsorption of UVS-NZVI to $Pb^{2+}$ was still more consistent with Langmuir model compared to Freundlich. In addition, the Freundlich constant $n_f$ for single and mixed adsorption experiment were all between 0.1 and 1, indicating that the adsorption process of $Pb^{2+}$ was favourable [55,56].

The maximum adsorption capacity of UVS-NZVI for $Cd^{2+}$ was 46.07 mg g$^{-1}$ in the solution containing only $Cd^{2+}$. Because the correlation coefficient of Langmuir model (0.9797) fitting is greater than that of Freundlich model (0.9344), the $Cd^{2+}$ adsorption by UVS-NZVI was more in line with the Langmuir model. Nevertheless, the presence of $Pb^{2+}$ observably decreased the Langmuir adsorption capacity of $Cd^{2+}$ to 3.31 mg g$^{-1}$ (figure 15*b*), which indicates the competition for active sites between them. The compound Pb-hydroxide on the surface of UVS-NZVI inhibited the formation of Cd–Fe-hydroxide in mixed solution and the surface cation exchange sites were completely occupied by Pb-hydroxide precipitation and Pb [57–59]. Hence, there was almost no $Cd^{2+}$ adsorbed on the UVS-NZVI composite. In summary, $Pb^{2+}$ is far more competitive than $Cd^{2+}$.

In separate adsorption experiment for $Pb^{2+}$ or $Cd^{2+}$, the correlation coefficients of Langmuir fitting are 0.9911 and 0.9797, respectively, which are higher than the correlation coefficients of Freundlich fitting (0.9234 and 0.9344). Since the important assumption of the Langmuir model is monolayer adsorption, and typical chemical adsorption is also monolayer adsorption, it can be inferred that the adsorption of $Pb^{2+}$ or $Cd^{2+}$ by UVS-NZVI may be chemical adsorption.

Since the activated shell (UVS) has a porous structure and a larger specific surface area than other biomass materials, the adsorption capacity of UVS-NZVI has certain advantages. The maximum

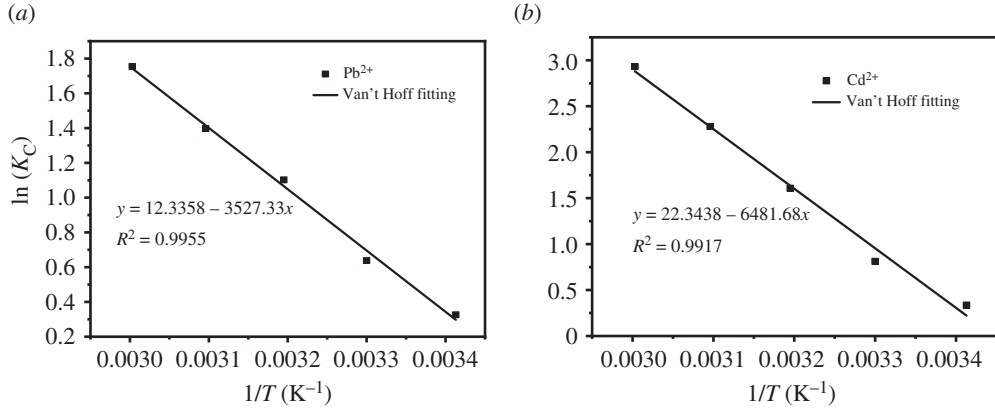

**Figure 16.** Fitting results of $K_c$ and $1/T$ of $Pb^{2+}$ and $Cd^{2+}$.

**Table 2.** Maximum adsorption capacity of different biomass materials.

| ion species | biomass carrier | maximum adsorption capacity (mg g$^{-1}$) |
|---|---|---|
| $Pb^{2+}$ | UVS | 93.01 |
| | *Posidonia oceanica* seaweed | 49.63 |
| | cassava fibre | 52.97 |
| | sepiolite | 33.42 |
| $Cd^{2+}$ | UVS | 46.07 |
| | *Posidonia oceanica* seaweed | 33.15 |
| | sugarcane fibre | 45.36 |
| | chitosan | 19.13 |

**Table 3.** Adsorption thermodynamic parameters of $Pb^{2+}$ and $Cd^{2+}$.

| ion species | $T$ (K) | $\Delta G$ (kJ mol$^{-1}$) | $\Delta H$ (kJ mol$^{-1}$) | $\Delta S$ (J mol$^{-1}$ K$^{-1}$) | $R^2$ |
|---|---|---|---|---|---|
| $Pb^{2+}$ | 293.15 | −0.74 | 29.33 | 102.56 | 0.9955 |
| | 303.15 | −1.76 | | | |
| | 313.15 | −2.79 | | | |
| | 323.15 | −3.81 | | | |
| | 333.15 | −4.84 | | | |
| $Cd^{2+}$ | 293.15 | −0.57 | 53.89 | 185.77 | 0.9917 |
| | 303.15 | −2.43 | | | |
| | 313.15 | −4.82 | | | |
| | 323.15 | −6.14 | | | |
| | 333.15 | −7.99 | | | |

adsorption capacities of $Pb^{2+}$ and $Cd^{2+}$ by nano-zero-valent iron supported by different biomass carriers are listed in table 2 [7,60–63].

Van't Hoff's fitting results and parameters for temperature changes are shown in figure 16 and table 3. According to the calculation results of thermodynamic parameters, the $\Delta H$ of UVS-NZVI for $Pb^{2+}$ and $Cd^{2+}$ adsorption are 29.33 and 53.89 kJ mol$^{-1}$, respectively, which means that the adsorption processes of UVS-NZVI for $Pb^{2+}$ and $Cd^{2+}$ are endothermic. According to the study of Ma *et al.* [64], the enthalpy change range of physical adsorption is between 2.10 and 20.90 kJ mol$^{-1}$, and the enthalpy change range of chemical adsorption is between 20.90 and 418.40 kJ mol$^{-1}$. Therefore, the

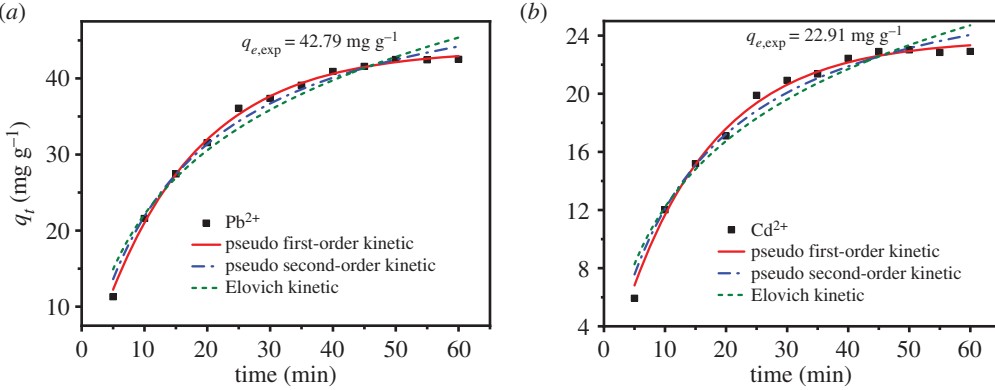

**Figure 17.** Adsorption kinetic model for $Pb^{2+}$ (*a*) and $Cd^{2+}$ (*b*) ions on UVS-NZVI.

**Table 4.** Adsorption kinetic parameters for $Pb^{2+}$ and $Cd^{2+}$ ions on UVS-NZVI.

| adsorption kinetic model | parameters | metal ions | |
|---|---|---|---|
| | | $Pb^{2+}$ | $Cd^{2+}$ |
| pseudo first-order | $q_e$ (mg g$^{-1}$) | 43.79 | 23.75 |
| | $k_1$ (min$^{-1}$) | 0.0655 | 0.0676 |
| | $R^2$ | 0.9976 | 0.9938 |
| pseudo second-order | $q_e$ (mg g$^{-1}$) | 55.56 | 29.99 |
| | $k_2$ (min$^{-1}$) | 0.0012 | 0.0023 |
| | $R^2$ | 0.9874 | 0.9782 |
| Elovich | $\alpha_E$ (mg (g min)$^{-1}$) | 0.0259 | 0.0551 |
| | $\beta_E$ (g mg$^{-1}$) | 13.6274 | 6.8208 |
| | $R^2$ | 0.9616 | 0.9506 |

adsorption of $Pb^{2+}$ and $Cd^{2+}$ by UVS-NZVI is chemical adsorption in this study, which is consistent with the results obtained by Langmuir model fitting. In the temperature range of 293.15–333.15 K, the $\Delta G$ range of UVS-NZVI for $Pb^{2+}$ adsorption is between −4.84 and −0.74 kJ mol$^{-1}$, and the $\Delta G$ range for $Cd^{2+}$ adsorption is between −7.99 and −0.57 kJ mol$^{-1}$. With the increase of temperature, the two $\Delta G$ values both decrease, indicating that the adsorption process is spontaneous.

The calculation results of the above thermodynamic parameters fully prove that the adsorption process of $Pb^{2+}$ and $Cd^{2+}$ by UVS-NZVI is a spontaneous chemical adsorption process.

### 3.2.5. Adsorption kinetics analysis

The effect of adsorption time for the removal of $Pb^{2+}$ and $Cd^{2+}$ ions by the UVS-NZVI was studied with ion concentration = 50 mg l$^{-1}$, and the results are shown in figure 17*a*,*b*. It was observed that the adsorption amount of metal ions was increased with an increase of the time in the range of 5–60 min whether it was $Pb^{2+}$ or $Cd^{2+}$, and the adsorption amount was increased quickly in the first 30 min, then increased gradually. As a result, $Pb^{2+}$ reached adsorption equilibrium on the UVS-NZVI at 50 min and $Cd^{2+}$ at 45 min. The adsorption of ions increased rapidly at the initial stage due to the accessibility of huge amount of binding sites on the surface of the UVS-NZVI. Thus the percentage removal of ions significantly increased at lesser adsorption time. At higher adsorption time, the rate of adsorption increased slowly until equilibrium due to the available binding sites being gradually decreased; for that reason, it took a long time for the adsorption process to reach equilibrium [18].

The adsorption mechanism of the $Pb^{2+}$ and $Cd^{2+}$ ions on UVS-NZVI was analysed by using three adsorption kinetic models, namely pseudo first-order kinetic model (the reaction rate is linearly related to the concentration of a reactant, and this model is based on the fact that rate-determining step is a physical process), pseudo second-order kinetic model (the reaction rate is linearly related to

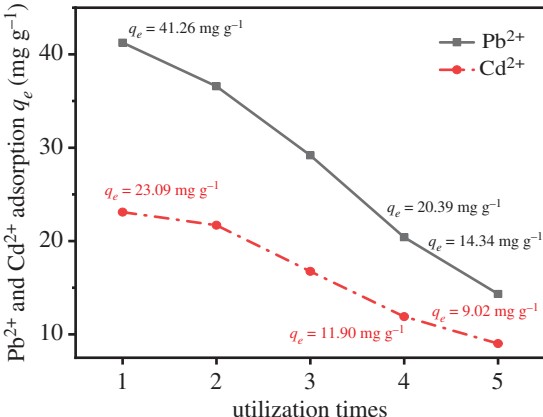

**Figure 18.** The effect of utilization times on adsorption of $Pb^{2+}$ and $Cd^{2+}$.

the concentration of two reactants, and this model is based on the fact that rate-determining step is a chemical reaction) and Elovich kinetic model (the Elovich model is suitable for processes with irregular data or with large activation energy) [65]. The fitting results are shown in figure 17a,b, and the kinetic parameters for the adsorption of $Cd^{2+}$ and $Pb^{2+}$ are listed in table 4. The fitted adsorption capacity ($q_e$) was compared with the equilibrium adsorption amount ($q_{e,\text{exp}}$) in the experiment. It can be observed from table 4 that the $q_e$ of pseudo first-order kinetic for the adsorption of $Cd^{2+}$ and $Pb^{2+}$ by UVS-NZVI was much closer to the $q_{e,\text{exp}}$ compared with other kinetic models. In addition, the determination coefficient of the pseudo first-order kinetic was higher than that of pseudo second-order and Elovich kinetic model for both $Cd^{2+}$ and $Pb^{2+}$, which indicates that the physical adsorption is the rate-determining step [66]. It may be possible that the huge specific surface area of UVS material increases the contact area significantly between NZVI and the solution compared with other carrier materials and this advantage plays a decisive role in the adsorption process, which agrees with Huang *et al.* [67].

### 3.2.6. Recyclability analysis

Figure 18 shows the adsorption amount variation trend of multiple adsorption of $Pb^{2+}$ or $Cd^{2+}$ by UVS-NZVI in aqueous solution. After five recycles, the adsorption of $Pb^{2+}$ decreased from the initial 41.26 to 14.34 mg g$^{-1}$ and the $Cd^{2+}$ from 23.09 to 9.02 mg g$^{-1}$. The reason was that the NZVI particles were exhausted by chemical reduction and the absorbed $Pb^{2+}$ and $Cd^{2+}$ ions were not completely resolved [42]. In adsorption of the fourth recycle, the UVS-NZVI adsorption amount of $Pb^{2+}$ was 20.39 mg g$^{-1}$ and the $Cd^{2+}$ adsorption amount was 11.90 mg g$^{-1}$, which indicates that UVS-NZVI can still achieve about 50% effect of the initial adsorption after four recycles.

## 4. Conclusion

In summary, UVS material-supported NZVI was successfully synthesized. NZVI particles can be evenly distributed on the porous layered structured UVS surface, which can effectively prevent NZVI from reuniting. A large number of adsorption sites can be derived from a large amount of $Fe^0$, a small amount of $Fe_2O_3$, $Fe(OH)_3$ and FeOOH on the surface of UVS-NZVI. The adsorption amounts of $Pb^{2+}$ and $Cd^{2+}$ both increased with the initial pH and then remained unchanged, and the adsorption effect of the two ions reached the peak at pH = 6.00 and the experimental maximum adsorption capacities of $Pb^{2+}$ and $Cd^{2+}$ were 29.91 and 38.99 mg g$^{-1}$, respectively. The presence of $Cu^{2+}$ has a significant effect on the adsorption of $Pb^{2+}$ and $Cd^{2+}$. $Ni^{2+}$ has a great influence on the adsorption of $Cd^{2+}$, but there is almost no adsorption competition with $Pb^{2+}$. Thermodynamic studies showed that the adsorption of $Pb^{2+}$ and $Cd^{2+}$ by UVS-NZVI is a chemical adsorption. Kinetic studies demonstrated that the physical adsorption was the rate-determining step. Further, the recyclability experiment suggested that UVS-NZVI had a good recycling effect in treating $Pb^{2+}$ and $Cd^{2+}$ in wastewater. Overall, this study not only provides an effective application approach for shellfish biomass resources, but also promotes the wider application of NZVI adsorbent, which makes UVS-NZVI have certain application potential in the heavy metal ions adsorption field.

Data accessibility. The data supporting the findings of this study are available at the Dryad Digital Repository: https://doi.org/10.5061/dryad.gqnk98sjz [68].

Authors' contributions. Z.W., X.W., S.L., Y.W., Z.T. and Q.D. undertook the research. Z.W. and X.W. participated in every part of the experiment and wrote the manuscript, they should be considered co-first authors.

Competing interests. We declare we have no competing interests.

Funding. This research was supported by the Natural Science Foundation of China (grant no. 21767008) and the Natural Science Foundation of Hainan Province (grant no. 219000).

Acknowledgements. This work was financially supported by the China and Key Laboratory of Ministry of Education of Advanced Materials of Tropical Island Resources (Hainan University) (AM2017-11).

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
