## [Reviewer comments · Royal Society Open Science]

Review History

RSOS-201192.R0 (Original submission)

Review form: Reviewer 1 (Ala Abdessemed)

Is the manuscript scientifically sound in its present form?

Yes

Are the interpretations and conclusions justified by the results?

Yes

Is the language acceptable?

Yes

Do you have any ethical concerns with this paper?

No

Have you any concerns about statistical analyses in this paper?

No

Recommendation?

Major revision is needed (please make suggestions in comments)

Comments to the Author(s)

This manuscript deals with Biomass material – undulating venus shell – supported nanoscale zero-valent iron adsorbing the Pb²⁺ and Cd²⁺ ions in aqueous solution. This study suggested the use of a composite material for adsorption properties for Pb²⁺ and Cd²⁺ in water and the thermodynamic and kinetic mechanism.

I have some comments/suggestions:

- Keywords should be reviewed (Pb²⁺ and Cd²⁺ which is very important in the study is not mentioned in the keywords),
- It should be said why the high temperature calcination activation was chosen (why not using chemical activation?),
- Is your method of preparation is low cost and green? If yes, please justify.
- BET analysis will be helpful to understand the role of surface area on adsorption studies. The authors should add it into the revised manuscript,
- On what basis you have chosen 0.04 g of UVS-NZVI to use it in the adsorption experiments.
- Why did you choose a time of 2 hours for the adsorption study? Is it the time necessary to achieve equilibrium?
- What is the analytical method used for the determination of the residual concentrations of Pb²⁺ and Cd²⁺ ? (it should be add in the experiment section),
- Why in the kinetics experiments you changed the adsorption time from 2 h to 60 min and the initial concentration used from 20 mg.l⁻¹ to 50 mg.l⁻¹ and the adsorbent mass from 0.04 g to 0.02 g?
- Please check and complete conditions used in figure captions.

In conclusion and in my opinion, currently, this work can be recommended for publication after addressing the different issues.

Review form: Reviewer 2 (Ismat Ali)

Is the manuscript scientifically sound in its present form?

Yes

Are the interpretations and conclusions justified by the results?

Yes

Is the language acceptable?

No

Do you have any ethical concerns with this paper?

No

Have you any concerns about statistical analyses in this paper?

No

Recommendation?

Major revision is needed (please make suggestions in comments)

Comments to the Author(s)

General observations

- 1) The subject of the paper is interesting but the novelty aspect is not highlighted enough.
- 2) The title of this manuscript should be changed to be more simple and informative.
- 3) The influence of ionic strength and the presence of co-existing ions need to be studied.
- 4) BET surface area is very important to an adsorbent, which should be provided.

- 5) Did the authors repeat the experiments? If yes, the average values and errors should be discussed.
- 6) One of the most important parameters is temperature effect. Authors should study the effect of temperature, calculate, report and discuss the thermodynamic parameters.
- 7) It will be better if authors add a table that involves the maximum adsorption capacity for various biomass materials used to adsorb Pb^{2+} and Cd^{2+} ions. This helps readers to realize the importance of this work.
- 8) Compressive revision for English language is required.
- 9) The nature of adsorption (Physical or chemical) should be mentioned with evidences.

Page 2, line 32: the maximum adsorption capacity should be mentioned. It should be calculated from the well fitted isotherm model.

Page 3, line 50: how did authors adjust the required mass of adsorbent? i.e authors should perform experiments to determine the optimal mass of adsorbent.

Page 3, Line 55: these reagents instead of above reagents

Page 4, line 45: Throughout the manuscript, please don't use (above) or (below) to mention equations. Instead use the equation number.

Page 4, line 53: (Pb^{2+} precipitated above pH 7.0). Please cite a reference to support this statement.

Page 5, line 33: were adjusted to pH 6.0. (no need to use the word respectively)

Page 10, line 3: the results given for adsorption isotherm are - to some extent - confusing. Which model is obeyed (for Cd^{2+}).

Decision letter (RSOS-201192.R0)

Dear Dr Luo:

Title: Biomass material - undulating venus shell - supported nanoscale zero-valent iron adsorbing the Pb^{2+} and Cd^{2+} ions in aqueous solution

Manuscript ID: RSOS-201192

The editor assigned to your manuscript has now received comments from reviewers. We would like you to revise your paper in accordance with the referee and Subject Editor suggestions which can be found below (not including confidential reports to the Editor). Please note this decision does not guarantee eventual acceptance.

Please submit your revised paper before 27-Aug-2020. Please note that the revision deadline will expire at 00.00am on this date. If we do not hear from you within this time then it will be assumed that the paper has been withdrawn. In exceptional circumstances, extensions may be possible if agreed with the Editorial Office in advance. We do not allow multiple rounds of revision so we urge you to make every effort to fully address all of the comments at this stage. If deemed necessary by the Editors, your manuscript will be sent back to one or more of the original reviewers for assessment. If the original reviewers are not available we may invite new reviewers.

To revise your manuscript, log into <http://mc.manuscriptcentral.com/rsos> and enter your Author Centre, where you will find your manuscript title listed under "Manuscripts with Decisions." Under "Actions," click on "Create a Revision." Your manuscript number has been

appended to denote a revision. Revise your manuscript and upload a new version through your Author Centre.

RSC Associate Editor:
Comments to the Author:
(There are no comments.)

RSC Subject Editor:
Comments to the Author:
(There are no comments.)

Reviewers' Comments to Author:

Reviewer: 1

Comments to the Author(s)

This manuscript deals with Biomass material – undulating venus shell – supported nanoscale zero-valent iron adsorbing the Pb²⁺ and Cd²⁺ ions in aqueous solution. This study suggested the use of a composite material for adsorption properties for Pb²⁺ and Cd²⁺ in water and the thermodynamic and kinetic mechanism.

I have some comments/suggestions:

- Keywords should be reviewed (Pb²⁺ and Cd²⁺ which is very important in the study is not mentioned in the keywords),
- It should be said why the high temperature calcination activation was chosen (why not using chemical activation?),
- Is your method of preparation is low cost and green? If yes, please justify.
- BET analysis will be helpful to understand the role of surface area on adsorption studies. The authors should add it into the revised manuscript,
- On what basis you have chosen 0.04 g of UVS-NZVI to use it in the adsorption experiments.
- Why did you choose a time of 2 hours for the adsorption study? Is it the time necessary to achieve equilibrium?

- What is the analytical method used for the determination of the residual concentrations of Pb^{2+} and Cd^{2+} ? (it should be add in the experiment section),
- Why in the kinetics experiments you changed the adsorption time from 2 h to 60 min and the initial concentration used from 20 mg.l⁻¹ to 50 mg.l⁻¹ and the adsorbent mass from 0.04 g to 0.02 g?
- Please check and complete conditions used in figure captions.

In conclusion and in my opinion, currently, this work can be recommended for publication after addressing the different issues.

Reviewer: 2

Comments to the Author(s)

General observations

- 1) The subject of the paper is interesting but the novelty aspect is not highlighted enough.
- 2) The title of this manuscript should be changed to be more simple and informative.
- 3) The influence of ionic strength and the presence of co-existing ions need to be studied.
- 4) BET surface area is very important to an adsorbent, which should be provided.
- 5) Did the authors repeat the experiments? If yes, the average values and errors should be discussed.
- 6) One of the most important parameters is temperature effect. Authors should study the effect of temperature , calculate, report and discuss the thermodynamic parameters.
- 7) It will be better if authors add a table that involves the maximum adsorption capacity for various biomass materials used to adsorb Pb^{2+} and Cd^{2+} ions. This helps readers to realize the importance of this work.
- 8) Compressive revision for English language is required.
- 9) The nature of adsorption (Physical or chemical) should be mentioned with evidences.

Page 2, line 32: the maximum adsorption capacity should be mentioned. It should be calculated from the well fitted isotherm model.

Page 3, line 50: how did authors adjust the required mass of adsorbent? i.e authors should perform experiments to determine the optimal mass of adsorbent.

Page 3, Line 55: these reagents instead of above reagents

Page 4, line 45: Throughout the manuscript, please don't use (above) or (below) to mention equations. Instead use the equation number.

Page 4, line 53: (Pb^{2+} precipitated above pH 7.0). Please cite a reference to support this statement.

Page 5, line 33: were adjusted to pH 6.0. (no need to use the word respectively)

Page 10, line 3: the results given for adsorption isotherm are - to some extent - confusing. Which model is obeyed (for Cd^{2+}).

Author's Response to Decision Letter for (RSOS-201192.R0)

See Appendix A.

RSOS-201192.R1 (Revision)

Review form: Reviewer 1 (Ala Abdessemed)

Is the manuscript scientifically sound in its present form?

Yes

Are the interpretations and conclusions justified by the results?

Yes

Is the language acceptable?

Yes

Do you have any ethical concerns with this paper?

No

Have you any concerns about statistical analyses in this paper?

No

Recommendation?

Accept as is

Comments to the Author(s)

After revision the authors take all comments into consideration. I believe that the manuscript in its revised form can be accepted for publication.

Review form: Reviewer 2 (Ismat Ali)

Is the manuscript scientifically sound in its present form?

Yes

Are the interpretations and conclusions justified by the results?

Yes

Is the language acceptable?

Yes

Do you have any ethical concerns with this paper?

No

Have you any concerns about statistical analyses in this paper?

No

Recommendation?

Accept with minor revision (please list in comments)

Comments to the Author(s)

Authors have performed carefully all the required correction

- 1) Affiliation 1: I guess chemical NOT chencial
- 2) Spaces between words and symbols e.g. page 11 line 28 ($\text{pH} \geq 5.5$). this should corrected throughout the manuscript.
- 3) Consistent decimals should be used throughout the manuscript. For pH values, please always use TWO DECIMALS e.g. page 11 line 57 should be ($\text{pH} = 6.00$)
- 4) Page 5 line 8: please add your statement (Reply: At $30\text{ }^\circ\text{C}$, the solubility product of $\text{Pb}(\text{OH})_2$ is about 1×10^{-17} . According to the concentration of Pb^{2+} in the solution, calculations show that when the pH of the solution is slightly greater than 7, Pb^{2+} begins to precipitate.) to the text.

Decision letter (RSOS-201192.R1)

Dear Dr Luo:

Title: Shell biomass material supported nano-zero valent iron to remove Pb²⁺ and Cd²⁺ in water
Manuscript ID: RSOS-201192.R1

Thank you for submitting the above manuscript to Royal Society Open Science. On behalf of the Editors and the Royal Society of Chemistry, I am pleased to inform you that your manuscript will be accepted for publication in Royal Society Open Science subject to minor revision in accordance with the referee suggestions. Please find the reviewers' comments at the end of this email.

The reviewers and handling editors have recommended publication, but also suggest some minor revisions to your manuscript. Therefore, I invite you to respond to the comments and revise your manuscript.

Because the schedule for publication is very tight, it is a condition of publication that you submit the revised version of your manuscript before 20-Sep-2020. Please note that the revision deadline will expire at 00.00am on this date. If you do not think you will be able to meet this date please let me know immediately.

Kind regards,
Dr Laura Smith
Publishing Editor, Journals

RSC Associate Editor:
Comments to the Author:
(There are no comments.)

RSC Subject Editor:
Comments to the Author:
(There are no comments.)

Reviewer comments to Author:
Reviewer: 2

Comments to the Author(s)

Authors have performed carefully all the required correction

- 1) Affiliation 1: I guess chemical NOT chemical
- 2) Spaces between words and symbols e.g. page 11 line 28 ($\text{pH} \geq 5.5$). this should corrected throughout the manuscript.
- 3) Consistent decimals should be used throughout the manuscript. For pH values, please always use TWO DECIMALS e.g. page 11 line 57 should be ($\text{pH} = 6.00$)
- 4) Page 5 line 8: please add your statement (Reply: At 30 °C, the solubility product of $\text{Pb}(\text{OH})_2$ is about 1×10^{-17} . According to the concentration of Pb^{2+} in the solution, calculations show that when the pH of the solution is slightly greater than 7, Pb^{2+} begins to precipitate.) to the text.

Reviewer: 1

Comments to the Author(s)

After revision the authors take all comments into consideration. I believe that the manuscript in its revised form can be accepted for publication.

Author's Response to Decision Letter for (RSOS-201192.R1)

See Appendix B.

Decision letter (RSOS-201192.R2)

Dear Dr Luo:

Title: Shell biomass material supported nano-zero valent iron to remove Pb²⁺ and Cd²⁺ in water
Manuscript ID: RSOS-201192.R2

It is a pleasure to accept your manuscript in its current form for publication in Royal Society Open Science. The chemistry content of Royal Society Open Science is published in collaboration with the Royal Society of Chemistry.

RSC Associate Editor
Comments to the Author:
(There are no comments.)

Reviewer(s)' Comments to Author:

Appendix A

Dear editor:

Thank you for taking the time to review the manuscript and make suggestions to the author. Based on the questions and suggestions raised by the two reviewers, We responded to the reviewers' questions point-to-point and revised the manuscript in detail. We have attached replies to the reviewer's comments in the following article.

Thank you very much for your attention and consideration.

Best regards,

Sincerely yours,

Shengxu Luo

Response to reviewer 1

Dear reviewer:

Thank you for reviewing the manuscript during your busy schedule. Based on your suggestions, we have made detailed revisions to the manuscript and answered your questions in detail.

Question 1: Keywords should be reviewed (Pb²⁺ and Cd²⁺ which is very important in the study is not mentioned in the keywords),

Reply: The supported adsorbent synthesized in this study is mainly used to explore its adsorption effect on Pb²⁺ and Cd²⁺ in water. According to your suggestion, we have added Pb²⁺ and Cd²⁺ to the keywords. The modified keywords are: UVS-NZVI, adsorption, isotherm, kinetic, recyclability, Pb²⁺, Cd²⁺.

Question 2: It should be said why the high temperature calcination activation was chosen (why not using chemical activation?),

Reply: The organic matter in the shell components is oxidized and removed under high temperature calcination. The calcined shell forms a porous structure with an increased specific surface area. If chemical activation is used, although strong acid conditions can remove organic matter, it will destroy the CaCO₃ skeleton structure of the shell. However, the removal of organic matter under alkaline conditions is not complete. Based on your suggestion, we have added the explanation in the fourth paragraph of the "Introduction"(Page 2).

Question 3: Is your method of preparation is low cost and green? If yes, please justify.

Reply: The preparation method used is low-cost and green. The shell substrate used in the research is reused from waste shells. The reducing agent NaBH₄ is one of the most commonly used low-cost environmentally friendly reducing agents, and the excess reducing agent can be removed with a little dilute acid.

Question 4: BET analysis will be helpful to understand the role of surface area on adsorption studies. The authors should add it into the revised manuscript,

Reply: Based on your suggestion, we have added BET analysis to the article(3.1 Characterization of adsorbents, Page 9), and the details are as follows:

Nitrogen sorption isotherm and pore size distribution are shown in Fig. 9 and Fig. 10. The specific surface area and pore diameter of UVS-NZVI calculated by Brunauer-Emmett-Teller method and Barret-Joyner-Halenda method are 83.770 m² • g⁻¹ and 3.537 nm respectively, which indicates that the porous structure of shell material makes nano-zero-valent iron have a larger contact area with solution, and the material may have higher adsorption performance.

Fig. 9 Nitrogen sorption isotherm of the sample recorded at 77 K.

Fig. 10 Pore size distribution.

Question 5: On what basis you have chosen 0.04 g of UVS-NZVI to use it in the adsorption experiments.

Reply: Before starting the experiment, we have conducted a preliminary experiment to determine the optimal amount of adsorbent. Based on your suggestion, we have supplemented this part of the experiment in the manuscript(3.2.1 Influence of adsorbent dosage on the adsorption of Pb^{2+} and Cd^{2+} ions, Page 10), and the details are as follows:

The effects of different adsorbent dosage on the adsorption of Pb^{2+} and Cd^{2+} by UVS-NZVI are shown in Fig. 11. For Pb^{2+} and Cd^{2+} , when the amount of adsorbent is less than 0.04 g, the adsorption capacity remains almost unchanged. When the amount of adsorbent is greater than 0.04 g, the adsorption capacity drops significantly. This may be due to the ion concentration is too small, the added adsorbent has not reached its optimal adsorption effect. Therefore, in order to achieve the best removal effect and the best utilization of the adsorbent, the amount of adsorbent in subsequent experiments will be 0.04 g.

Fig. 11 The effect of adsorbent dosage on adsorption of Pb²⁺ and Cd²⁺.

Question 6: Why did you choose a time of 2 hours for the adsorption study? Is it the time necessary to achieve equilibrium?

Reply: According to the investigation of the reaction in the preliminary experiment process and the subsequent kinetic experiment process, the adsorption reaction can reach the adsorption equilibrium in about 60 minutes. Therefore, choosing 2 h for the adsorption study can reach the necessary time for equilibrium.

Question 7: What is the analytical method used for the determination of the residual concentrations of Pb²⁺ and Cd²⁺ ? (it should be add in the experiment section),

Reply: The concentration of Pb²⁺ and Cd²⁺ is measured by flame atomic absorption spectrophotometer. According to your suggestion, we have added the measured standard curve parameters to the manuscript(2.5 Calculation method for adsorption amount, Page 4), and the details are as follows:

Prepare lead standard series solutions with mass concentrations of 5 mg · L⁻¹, 10 mg · L⁻¹, 20 mg · L⁻¹, 30 mg · L⁻¹, and 40 mg · L⁻¹ with 0.15 mg · L⁻¹ dilute nitric acid solution. Measure the absorbance of the standard series of solutions by flame atomic absorption method, draw the standard curve of A_{Pb} (absorbance) - C_{Pb} (mass concentration of Pb²⁺). Measurement conditions: gas flow rate 1500 mL · min⁻¹, height 5.0 mm, position -0.5 mm, working lamp current 2.0 mA, spectral bandwidth 0.4 nm, negative high voltage 300 V.

The standard curve of Pb²⁺ is shown in Fig. 1, the standard curve equation: $y=0.00906+ 0.02038x$, $R^2=0.99983$. Within the range of Pb²⁺ mass concentration of 5-40 mg · L⁻¹, the linear relationship is well.

Fig. 1 Standard curve for determining Pb^{2+} content

Prepare cadmium standard series solutions with mass concentrations of $0 \text{ mg} \cdot \text{L}^{-1}$, $1 \text{ mg} \cdot \text{L}^{-1}$, $2 \text{ mg} \cdot \text{L}^{-1}$, $3 \text{ mg} \cdot \text{L}^{-1}$, $4 \text{ mg} \cdot \text{L}^{-1}$ and $5 \text{ mg} \cdot \text{L}^{-1}$ with $0.15 \text{ mg} \cdot \text{L}^{-1}$ dilute nitric acid solution. Measure the absorbance of the standard series of solutions by flame atomic absorption method, draw the standard curve of A_{Cd} (absorbance) - C_{Cd} (mass concentration of Cd^{2+}). Measurement conditions: gas flow rate $1300 \text{ mL} \cdot \text{min}^{-1}$, height 5.0 mm , position -0.5 mm , working lamp current 2.0 mA , spectral bandwidth 0.4 nm , negative high voltage 300 V .

The standard curve of Cd^{2+} is shown in Fig. 2, the standard curve equation: $y=0.01471+0.13171x$, $R^2=0.9968$. Within the range of Cd^{2+} mass concentration of $0\text{-}5 \text{ mg} \cdot \text{L}^{-1}$, the linear relationship is well.

Fig. 2 Standard curve for determining Cd^{2+} content

Question 8: Why in the kinetics experiments you changed the adsorption time from 2 h to 60 min and the initial concentration used from $20 \text{ mg} \cdot \text{L}^{-1}$ to $50 \text{ mg} \cdot \text{L}^{-1}$ and the adsorbent mass from 0.04 g to 0.02 g ?

Reply: According to the results of single factor experiment and adsorption thermodynamic experiment, the synthesized supported adsorbent has good adsorption capacity (the maximum adsorption capacity for Pb^{2+} and Cd^{2+} reach $93.01 \text{ mg} \cdot \text{g}^{-1}$ and $46.07 \text{ mg} \cdot \text{g}^{-1}$ respectively) and adsorption efficiency (adsorption process can reach equilibrium in about 60 min). In order to reflect the high efficiency of the adsorbent, we reduced the amount of adsorbent (0.02 g) and increased the initial ion concentration ($50 \text{ mg} \cdot \text{L}^{-1}$) in the kinetic test. In the adjusted kinetic experiment, the adsorption of Pb^{2+} and Cd^{2+} by UVS-NZVI can reach equilibrium within 60 minutes.

Question 9: Please check and complete conditions used in figure captions.

Reply: Based on your suggestions, we have carefully checked and optimized the use of image captions in the text.

Thank you very much for your attention and consideration.

Response to reviewer 2

Dear reviewer:

Thank you for reviewing the manuscript during your busy schedule. Based on your suggestions, we have made detailed revisions to the manuscript and answered your questions in detail.

Question 1 : The subject of the paper is interesting but the novelty aspect is not highlighted enough.

Reply: In this study, pretreated shell materials were used to load nano-zero-valent iron. Compared with the previous matrix, the activated shell can not only support the nano-zero-valent iron and prevent agglomeration, but also make full use of the porous structure of the shell, increase the specific surface area of the adsorbent, and achieve physical adsorption and chemical adsorption to combine. Based on your suggestions, we have emphasized the above advantages in the manuscript(the third paragraph of the “Introduction”, Page 2).

Question 2 : The title of this manuscript should be changed to be more simple and informative.

Reply: Based on your suggestion, we have simplified the title of the article to “Shell biomass material supported nano-zero valent iron to remove Pb^{2+} and Cd^{2+} in water”.

Question 3: The influence of ionic strength and the presence of co-existing ions need to be studied.

Reply: The composition of industrial wastewater is complex. In addition to Pb^{2+} and Cd^{2+} studied in this experiment, it also contains a variety of heavy metal ions. According to your suggestions, we have conducted supplementary experiments to study the effects of common heavy metal ions Cu^{2+} and Ni^{2+} on the adsorption of Pb^{2+} and Cd^{2+} by UVS-NZVI, and add the experimental results and analysis to the manuscript(3.2.3 Influence of coexisting ion on the adsorption of Pb^{2+} and Cd^{2+} ions, Page 12), and the details are as follows:

The effects of different concentrations of Cu^{2+} and Ni^{2+} on the adsorption of Pb^{2+} and Cd^{2+} by UVS-NZVI are shown in Fig. 13 and Fig. 14.

According to Fig. 13, the presence of Cu^{2+} has a great influence on the adsorption of Pb^{2+} and Cd^{2+} . Since the oxidability of Cu^{2+} is similar to that of Pb^{2+} and slightly greater than that, as the concentration of Cu^{2+} increases, the adsorption performance of UVS-NZVI for Pb^{2+} decreases to a certain extent. The oxidability of Cu^{2+} is much greater than that of Cd^{2+} , so as the concentration of Cu^{2+} increases, the adsorption performance of UVS-NZVI on Pb^{2+} decreases significantly, and when the concentration of Cu^{2+} reaches $12 \text{ mg} \cdot \text{L}^{-1}$, the adsorption capacity of Cd^{2+} is only $1.931 \text{ mg} \cdot \text{g}^{-1}$, which shows that Cd^{2+} is hardly adsorbed.

The effects of Ni^{2+} on the adsorption of Pb^{2+} and Cd^{2+} by UVS-NZVI are shown in Fig. 14. Since the oxidability of Pb^{2+} is much greater than that of Ni^{2+} , with the increase of Ni^{2+} concentration, the adsorption capacity of Cd^{2+} hardly changes, and there is almost no adsorption competition between the two ions. The oxidability of Ni^{2+} is close to that of Cd^{2+} . As the concentration of Ni^{2+} increases, the adsorption capacity of Cd^{2+} decreases to a certain extent, but UVS-NZVI has still a significant adsorption capacity for Cd^{2+} .

Fig. 13 The influence of Cu²⁺ concentration on the adsorption of Pb²⁺ and Cd²⁺

Fig. 14 The influence of Ni²⁺ concentration on the adsorption of Pb²⁺ and Cd²⁺

Question 4: BET surface area is very important to an adsorbent, which should be provided.

Reply: Based on your suggestion, we have added the BET analysis to the manuscript(3.1 Characterization of adsorbents, Page 9), and the details are as follows:

Nitrogen sorption isotherm and pore size distribution are shown in Fig. 9 and Fig. 10. The specific surface area and pore diameter of UVS-NZVI calculated by Brunauer-Emmett-Teller method and Barret-Joyner-Halenda method are 83.770 m² · g⁻¹ and 3.537 nm respectively, which indicates that the porous structure of shell material makes nano-zero-valent iron have a larger contact area with solution, and the material may have higher adsorption performance.

Fig. 9 Nitrogen sorption isotherm of the sample recorded at 77 K.

Fig. 10 Pore size distribution.

Question 5: Did the authors repeat the experiments? If yes, the average values and errors should be discussed.

Reply: Due to the long period of the adsorption experiment, we did not repeat the experiment for the single factor conditions in the experiment. However, corresponding pre-experiments were carried out for different amounts of adsorbents and solutions of different concentrations. The adsorption effect under each condition of the pre-experiment was basically the same as in the experiment.

Question 6: One of the most important parameters is temperature effect. Authors should study the effect of temperature, calculate, report and discuss the thermodynamic parameters.

Reply: According to your suggestion, we have conducted supplementary experiments to study the effect of temperature on adsorption and discuss the thermodynamic parameters (3.2.4 Adsorption thermodynamic analysis, Page 14-15), and the details are as follows:

Van't Hoff's fitting results and parameters for temperature changes are shown in Fig. 16 and Table 3. According to the calculation results of thermodynamic parameters, the ΔH of UVS-NZVI for Pb^{2+} and Cd^{2+} adsorption are $29.33 \text{ kJ} \cdot \text{mol}^{-1}$ and $53.89 \text{ kJ} \cdot \text{mol}^{-1}$, respectively, which means that the adsorption processes of UVS-NZVI for Pb^{2+} and Cd^{2+} are endothermic. According to the study of Ma et al. [65], the enthalpy change range of physical adsorption is between $2.1\text{--}20.9 \text{ kJ} \cdot \text{mol}^{-1}$, and the enthalpy change range of chemical adsorption is between $20.9\text{--}418.4 \text{ kJ} \cdot \text{mol}^{-1}$. Therefore, the adsorption of Pb^{2+} and Cd^{2+} by UVS-NZVI is chemical adsorption in this study, which is consistent with the results obtained by Langmuir model fitting. In the temperature range of $293.15\text{K}\text{--}333.15\text{K}$, the ΔG range of UVS-NZVI for Pb^{2+} adsorption is between $-4.838\text{--}0.735 \text{ kJ} \cdot \text{mol}^{-1}$, and the ΔG range for Cd^{2+} adsorption is between $-7.99\text{--}0.568 \text{ kJ} \cdot \text{mol}^{-1}$. With the increase of temperature, the two ΔG both decreases, indicating that the adsorption process is spontaneous.

The calculation results of the above thermodynamic parameters fully prove that the adsorption process of Pb^{2+} and Cd^{2+} by UVS-NZVI is a spontaneous chemical adsorption process.

Fig. 16 Fitting results of K_c and $1/T$ of Pb^{2+} and Cd^{2+} .

Table 3 Adsorption thermodynamic parameters of Pb^{2+} and Cd^{2+}

Ion species	T(K)	$\Delta G(KJ \cdot mol^{-1})$	$\Delta H(KJ \cdot mol^{-1})$	$\Delta S(J \cdot mol^{-1} \cdot K^{-1})$	R^2
Pb^{2+}	293.15	-0.735			0.9955
	303.15	-1.761			
	313.15	-2.787	29.33	102.56	
	323.15	-3.812			
	333.15	-4.838			
Cd^{2+}	293.15	-0.568			0.9917
	303.15	-2.426			
	313.15	-4.824	53.89	185.77	
	323.15	-6.142			
	333.15	-7.999			

Question 7 : It will be better if authors add a table that involves the maximum adsorption capacity for various biomass materials used to adsorb Pb^{2+} and Cd^{2+} ions.

This helps readers to realize the importance of this work.

Reply: According to your suggestion, we have consulted relevant literature and compared the maximum adsorption capacity of Pb^{2+} and Cd^{2+} by different biomass materials(3.2.4 Adsorption thermodynamic analysis, Page 13-14), and the details are as follows:

Since the activated shell (UVS) has a porous structure and a larger specific surface area than other biomass materials, therefore, the adsorption capacity of UVS-NZVI has certain advantages. The maximum adsorption capacities of Pb^{2+} and Cd^{2+} by nano-zero-valent iron supported by different biomass carriers are listed in Table 2.[60-64]

Table 2 Maximum adsorption capacity of different biomass materials

Ion species	Biomass carrier	Maximum adsorption capacity(mg \cdot g $^{-1}$)
Pb^{2+}	UVS	93.01
	P. oceanica seaweed	49.63
	Cassava fiber	52.97
	sepiolite	33.42
Cd^{2+}	UVS	46.07
	P. oceanica seaweed	33.15
	Sugarcane fiber	45.36

Question 8: Compressive revision for English language is required.

Reply: According to your suggestion, we have revised the language expression in the text, corrected some grammatical errors, and performed language compression on some expressions.

Question 9: The nature of adsorption (Physical or chemical) should be mentioned with evidences.

Reply: According to the results of thermodynamic experiments, the adsorption of Pb^{2+} and Cd^{2+} by UVS-NZVI is chemical adsorption. Based on your suggestions, we have added the analysis results to the manuscript(3.2.4 Adsorption thermodynamic analysis, Page 13-14), and the details are as follows:

In separate adsorption experiment for Pb^{2+} or Cd^{2+} , The correlation coefficients of Langmuir fitting are 0.9911 and 0.9797 respectively, which are higher than the correlation coefficients of Freundlich fitting (0.9234 and 0.9344). Since the important assumption of the Langmuir model is monolayer adsorption, and typical chemical adsorption is also monolayer adsorption, it can be inferred that the adsorption of Pb^{2+} or Cd^{2+} by UVS-NZVI may be chemical adsorption.

According to the study of Ma et al.[65], the enthalpy change range of physical adsorption is between 2.1-20.9 $\text{KJ}\cdot\text{mol}^{-1}$, and the enthalpy change range of chemical adsorption is between 20.9-418.4 $\text{KJ}\cdot\text{mol}^{-1}$. Therefore, the adsorption of Pb^{2+} and Cd^{2+} by UVS-NZVI is chemical adsorption in this study, which is consistent with the results obtained by Langmuir model fitting.

Question: Page 2, line 32: the maximum adsorption capacity should be mentioned. It should be calculated from the well fitted isotherm model.

Reply: According to your suggestion, we have stated the maximum adsorption capacity in the text(3.2.4 Adsorption thermodynamic analysis, Page 13), the maximum adsorption capacity of UVS-NZVI for Pb^{2+} and Cd^{2+} are $93.01 \text{ mg} \cdot \text{g}^{-1}$ and $46.07 \text{ mg} \cdot \text{g}^{-1}$.

Question: Page 3, line 50: how did authors adjust the required mass of adsorbent? i.e authors should perform experiments to determine the optimal mass of adsorbent.

Reply: Before starting the experiment, we have conducted a preliminary experiment to determine the optimal amount of adsorbent. Based on your suggestion, we have supplemented this part of the experiment in the manuscript(3.2.1 Influence of adsorbent dosage on the adsorption of Pb^{2+} and Cd^{2+} ions, Page 10), and the details are as follows:

The effects of different adsorbent dosage on the adsorption of Pb^{2+} and Cd^{2+} by UVS-NZVI are shown in Fig. 11. For Pb^{2+} and Cd^{2+} , when the amount of adsorbent is less than 0.04 g, the adsorption capacity remains almost unchanged. When the amount of adsorbent is greater than 0.04 g, the adsorption capacity drops significantly. This may be due to the ion concentration is too small, the added adsorbent has not reached its optimal adsorption effect. Therefore, in order to achieve the best removal effect and the best utilization of the adsorbent, the amount of adsorbent in subsequent experiments will be 0.04 g.

Fig. 11 The effect of adsorbent dosage on adsorption of Pb²⁺ and Cd²⁺.

Question: Page 3, Line 55: these reagents instead of above reagents

Reply: According to your suggestion, we have optimized the expression (2.1 Materials, Page 3), and the details are as follows:

Absolute ethyl alcohol, FeCl₂·4H₂O (analytical grade), NaBH₄ (analytical grade), NaOH (analytical grade), HNO₃ (excellent grade), Cd(NO₃)₂ (analytical grade), Pb(NO₃)₂ (analytical grade), these reagents were purchased from Aladdin reagent (Shanghai) limited company.

Question : Page 4, line 45: Throughout the manuscript, please don't use (above) or (below) to mention equations. Instead use the equation number.

Reply: According to your suggestion, we have numbered the formulas in the manuscript in a uniform order and revised the expressions in the text.

Question : Page 4, line 53: (Pb²⁺ precipitated above pH 7.0). Please cite a reference to support this statement.

Reply: At 30 °C, the solubility product of Pb(OH)₂ is about 1*10⁻¹⁷. According to the concentration of Pb²⁺ in the solution, calculations show that when the pH of the solution is slightly greater than 7, Pb²⁺ begins to precipitate.

Question : Page 5, line 33: were adjusted to pH 6.0. (no need to use the word respectively)

Reply: According to your suggestion, we have deleted the repeated expressions.

Question: Page 10, line 3: the results given for adsorption isotherm are – to some extent – confusing. Which model is obeyed (for Cd²⁺).

Reply: According to your suggestions, we have re-expressed the analysis results (3.2.4 Adsorption thermodynamic analysis, Page 13), and the details are as follows:

The maximum adsorption capacity of UVS-NZVI for Cd²⁺ was 46.07 mg·g⁻¹ in the solution containing only Cd²⁺. Because the correlation coefficient of Langmuir model (0.9797) fitting is greater than that of Freundlich model (0.9344), the Cd²⁺ adsorption by UVS-NZVI was more in line with the Langmuir model.

Thank you very much for your attention and consideration.

Appendix B

Dear editor:

Thank you for taking the time to review the manuscript and make suggestions to the author. Based on the questions and suggestions raised by the reviewer 2, We responded to the questions point-to-point and revised the manuscript in detail. We have attached replies to the reviewer's comments in the following article.

Thank you very much for your attention and consideration.

Best regards,

Sincerely yours,

Shengxu Luo

Response to reviewer 2

Dear reviewer:

Thank you for reviewing the manuscript during your busy schedule. Based on your suggestions, we have made detailed revisions to the manuscript and answered your questions in detail.

Question 1: Affiliation 1: I guess chemical NOT chemical.

Reply: Thank you for pointing out our spelling mistakes, we have corrected the wrong word.

Question 2: Spaces between words and symbols e.g. page 11 line 28 ($\text{pH} \geq 5.5$). this should corrected throughout the manuscript..

Reply: According to your suggestion, we have checked the symbols in the full text and added spaces where necessary.

Question 3: Consistent decimals should be used throughout the manuscript. For pH values, please always use TWO DECIMALS e.g. page 11 line 57 should be ($\text{pH} = 6.00$).

Reply: According to your suggestion, we have unified the decimals in the text and corrected the decimal places of the pH value.

Question 4: Page 5 line 8: please add your statement (Reply: At $30\text{ }^{\circ}\text{C}$, the solubility product of $\text{Pb}(\text{OH})_2$ is about 1×10^{-17} . According to the concentration of Pb^{2+} in the solution, calculations show that when the pH of the solution is slightly greater than 7, Pb^{2+} begins to precipitate.) to the text.

Reply: According to your suggestion, we have added the corresponding statement to section 2.7 of the article.

Thank you very much for your attention and consideration.